# Liposomes as Tools to Improve Therapeutic Enzyme Performance

**DOI:** 10.3390/pharmaceutics14030531

**Published:** 2022-02-27

**Authors:** Maria Eugénia Meirinhos Cruz, Maria Luísa Corvo, Maria Bárbara Martins, Sandra Simões, Maria Manuela Gaspar

**Affiliations:** Research Institute for Medicines (iMed.ULisboa), Faculty of Pharmacy, Universidade de Lisboa, Av. Prof. Gama Pinto, 1649-003 Lisboa, Portugal; eugeniacruz@campus.ul.pt (M.E.M.C.); barbarafigueiramartins@edu.ulisboa.pt (M.B.M.)

**Keywords:** therapeutic enzymes, drug delivery systems, liposomes

## Abstract

The drugs concept has changed during the last few decades, meaning the acceptance of not only low molecular weight entities but also macromolecules as bioagent constituents of pharmaceutics. This has opened a new era for a different class of molecules, namely proteins in general and enzymes in particular. The use of enzymes as therapeutics has posed new challenges in terms of delivery and the need for appropriate carrier systems. In this review, we will focus on enzymes with therapeutic properties and their applications, listing some that reached the pharmaceutical market. Problems associated with their clinical use and nanotechnological strategies to solve some of their drawbacks (i.e., immunogenic reactions and low circulation time) will be addressed. Drug delivery systems will be discussed, with special attention being paid to liposomes, the most well-studied and suitable nanosystem for enzyme delivery in vivo. Examples of liposomal enzymatic formulations under development will be described and successful pre-clinical results of two enzymes, *L*-Asparaginase and Superoxide dismutase, following their association with liposomes will be extensively discussed.

## 1. Introduction

The so-called “change in paradigm” of molecules as drugs, from almost exclusively low molecular weight molecules to bioactive agents was due to research and biotechnological advances [1,2]. Nowadays, the use of biological molecules in research and in clinics, with an increasing interest over the years, is a reality. Some examples of those molecules used as bioactive agents include various types of proteins (i.e., enzymes, antigens, antibodies) and genetic material (i.e., DNA, mRNA, siRNA, oligonucleotides), amongst others. In this review, we will discuss enzymes as therapeutic agents and the interest on the construction of nanoformulations as an alternative to conventional formulations aiming to improve enzyme performance.

Enzymes are widely distributed in all animal and vegetal cells, playing a crucial role in all cellular metabolisms by being involved in critical reactions essential for the existence of life [3,4,5]. The absence or malfunctioning of one single enzyme can arise to illnesses which are commonly difficult to treat and control with conventional therapies. In fact, biopharmaceutics including enzymes, are the only treatment available to certain rare diseases, through enzymatic substitution or replacement [6].

Enzyme application as therapeutic agents, namely as digestive auxiliaries (amylases, proteases) has been reported since the 19th century [7,8]. Later, in the 1960s, it was understood that enzymes could constitute a promising class of bioagents for the treatment of several disorders, due to their almost unique properties resulting in high activity and selectivity. However, it is essential that native conformation is kept and that they are correctly formulated, stabilized and stored [9]. 

The enzymatic therapeutic strategy became more realistic after the establishment and increment of genetic engineering technologies that made possible the production of enzymes with the needed characteristics of purity, selectivity and amounts compatible with the clinical use [10]. The therapeutic potential of enzymes is very high, ranging from metabolic and inflammatory disorders, cardiac problems to even cancer [8,11]. The wide range of applications and the availability of enzymes with pharmaceutical characteristics justifies the rapidly emerging increase in the market mainly since 1987, after the first approval by the Food and Drug Administration (FDA) of recombinant tissue plasminogen activator, Alteplase, for the treatment of acute ischemic stroke [12]. However, irrespective of the enormous advantages of enzymes of high specificity and selectivity to the substrate, high activity and production facilities, their application for therapeutic purposes have been limited by several drawbacks, such as immunogenic reactions, low residence time in living organisms, rapid metabolization and/or degradation and loss of activity on storage. As an attempt to circumvent those limitations and improve therapeutic enzyme properties, strategies such as synthesis of enzyme derivatives, association to polymers, immobilization in inorganic supports or in drug delivery systems (DDS) have been developed over the years [8,13,14]. For the first time in 1972 and with this purpose, Gregory Gregoriadis and Brenda Ryman incorporated enzymes in lipid vesicles, liposomes, for treatment of lysosomal storage disease [15,16,17,18]. Since this pioneering work, liposomes and other DDS have been widely used to incorporate a wide variety of biomolecules, originating a new concept of nanoformulations and nanopharmaceutics, enriching the area of nanomedicines [19,20]. The rational for molecules association to DDS intends to improve the performance of the loaded material by several concomitant mechanisms, such as protection from degradation by the living organisms, change of pharmacokinetic profile and targeting the materials to specific sites, thus increasing the therapeutic outcome. After 50 years of investigation in this field, there are many cases of success, not only under the academic point of view but also in the industry, that have already commercialized a vast number of nanoformulations [21]. The most recent case of the DDS success, well-known worldwide, are the lipid nanoparticles incorporating genetic material and antigens that constitute the basis for the construction of vaccines against COVID-19 [17,22].

## 2. Enzymes with Therapeutic Properties

Enzymes, as a particular class of proteins, can be distinguished due to the specific binding and action on their targets with great affinity and specificity. They can be from vegetal, animal and antimicrobial origin [1,11,23]. Due to the advances in biotechnological methodologies (recombinant DNA), enzymes can be produced on a large scale without the need to be extracted from the natural sources and with advantages of higher purity and lower costs [9,24]. Due to the unique and extraordinary attributes of enzymes and the broad range of targets and possible action in the living organisms, they can present multiple applications for different pathologies. They range from mild digestive disorders to serious diseases such as cystic fibrosis, metabolic and cardiac disorders, inflammatory diseases and cancers, among others [8,12]. According to Drug Bank online [25] more than 100 enzymes evidencing therapeutic activity and acting by different mechanisms were identified. The purpose of the present work is not intended to be exhaustive in the nomination of therapeutic enzymes, as they can be found in recent revision publications [9,12].

Table 1 presents well-known types of enzymes, their action and application for selected diseases, which were chosen as examples. 

**Table 1 pharmaceutics-14-00531-t001:** Examples of enzyme applications against selected diseases. Adapted from de la Fuente et al. [12] and Kumari et al. [26].

Disorder	Enzymes	Mechanism of Action
Lysosomal and metabolic	galactosidase, beta—glucocerebrosidase, iduronate-2-sulfatase, α-L-iduronidase, arylsulphatase B Pancreatic enzymes, protin C, lactase, phenylalanine hydroxylase, adenosine deaminase, lysosomal acid lipase	Replacing those in failure or malfunction
Cancer	*L*-Asparaginase	Disrupting preferential substrate of leukemia cells
	arginine deaminase, kynureninase, yaluronidase	Destroying increased aminoacids in several cancers
	urate oxidase and rasburicase	Degrading uric acid overproduced in hyperuricemia in hepatocellular carcinoma, human colon cancer, leukemia, breast cancer and melanoma
Inflammation	superoxide dismutase, catalase	Scavenging oxygen radicals in arthritis and reperfusion ischemia, acting as cellular detoxification
Cardiovascular	urokinase, streptokinase, nattokinase	Degrading fibrin cloths into soluble fibrin
Digestive	proteases, lipases, amylases	Degrading food-derived subtracts

Table 2 lists some of the commercialized enzyme-based drugs. In view of the high number of enzymes identified as potential useful drugs and those already approved by the FDA and/or European Medicines Agency (EMA), it is difficult to understand the correlation with the low number of enzyme formulations that reached the market [12,27].

One of the reasons for that discrepancy is probably related to problems arising from the behavior of exogenous enzymes when administered in living organisms. In some cases, it can result in mild or even acute allergic reactions, including anaphylaxis and death [1,8,12]. Moreover, enzymes can be degraded during circulation in the organisms, with obvious loss of activity leading to a frequent administration regimen, and may present difficulties to reach the target sites, with increasing side effects. Some enzymes presenting in vitro activity in the native, free form have limited or inexistent activity after administration in humans being, in some cases, also associated with deleterious immunogenic reactions. This panorama can be changed if, alternatively to conventional pharmaceutical formulations of free enzymes, they would be associated with nano-DDS, able to improve their activity and in vivo safety [22].

## 3. Nano-Drug Delivery Systems as Tools for Improved Enzyme Delivery

Early efforts with intravenous administration of therapeutic enzymes based on passive uptake by target cells and tissues resulted in poor therapeutic activity. Improved activity, reduced immunogenicity and clearance were achieved with enzyme–nanocarrier association, the topic of extensive and fruitful research work [12,22]. Nanocarriers are usually defined as particles of the nanosize range; however, the mean diameter for therapeutic purposes can vary from 10–1000 nm [28]. Several nanocarrier systems were developed for drug delivery in general and therapeutic enzymes, in particular. The advantages of enzyme incorporation in such carriers are plenty recognized. Basically, the critical points are the preservation of enzyme catalytic activity, by maintaining their quaternary structure, while maximizing the encapsulation efficiency. Furthermore, the increase in enzyme stability in biological fluids enables enzyme availability to exert therapeutic effect [29]. This comprises a technological development based on the careful selection of carrier composition and assembly strategies to improve delivery and targeting of loaded enzymes [12]. Nanocarriers may possess different compositions, topographies and shapes. A large variety of nanostructures have been produced to be associated with macromolecules for therapeutic or diagnostic purposes, and the main developed categories are liposomes, nanoparticles [8], virosomes [30], extracellular vesicles [31] and erythrocytes [32]. The most widely used systems for enzyme incorporation are liposomes. They comprise phospholipid nanovesicles with a biocompatible structure, adaptable entrapment capabilities either for active or passive delivery of low and high molecular weight compounds. 

### Liposomes as Versatile Carriers for Enzyme Delivery

As one of the most-studied DDS of bioactive agents, liposomes have given rise to several commercialized formulations and many others in advanced stages of clinical trials. They present several advantages over other nanocarriers, such as structural versatility, biocompatibility, biodegradability, non-toxic and non-immunogenicity nature as well as the mild conditions for their preparation. They are spherical structures, consisting primarily of amphiphilic molecules (phospholipids), organized in bilayers and were the first lipid vesicles introduced by Bangham and Horne in 1964 [33]. The hydrophilic portions of lipids are aqueous oriented either on the outside of the vesicles or in the internal compartments. The hydrophobic portions are located within the bilayer without contacting water. The liposome architecture and stabilization are ensured by Van der walls forces that keep together the tails of the phospholipid’s nonpolar chains, and through hydrogen bonds between the water molecules and phospholipid’s polar heads [34]. Different types of liposomes have been described and several vesicle generations have been developed (Figure 1).

The physicochemical properties of liposomes are directly related to their lipid composition and preparation method. Lipid molecules such as phospholipids and cholesterol are the major components of the vesicular carriers. Their composition can be widely varied, through the choice and combination of their constituents, lipids and others. This allows the diversity of their physicochemical properties, depending on the following variables: charge of phospholipid polar head groups, saturation and length of the phospholipid acyl chains, presence of cholesterol or other non-lipid-charged molecules and proportion of the various constituents (lipidic or not). The environmental pH also influences drug–vesicle interactions [36]. The encapsulation of hydrophilic molecules which is dependent on the vesicle internal aqueous volume can be modulated through the preparation method chosen [36]. Moreover, the presence of cholesterol in the composition of vesicles made of fluid phospholipids reduces the permeability of lipid bilayers being the strategy often proposed to increase the encapsulation efficiency of hydrophilic drugs while keeping loaded material inside the vesicles [37]. Thus, it is possible to controllably prepare almost unlimited versions of liposomes and modulate their behavior in vivo, which is dependent on their characteristics.

Conventional liposomes (Figure 1A) fail to escape from capture by the mononuclear phagocytic system (MPS) after parenteral administration. Delivery strategies have been considered to circumvent their rapid clearance from the bloodstream. One of the most studied methodologies was accomplished by coating the liposome surface with polyethylene glycol (PEG) creating the so-called long-circulating liposomes, also named as “stealth liposomes” (Figure 1B) [38,39]. Other strategies consisted of the inclusion of different ligands at the liposome surface such as immunoglobulins, glycoproteins, transferrin, peptides, folate, etc., for preferentially targeting them to overexpressed receptors at affected sites [40]. Several other molecules, including enzymes can be covalently attached to PEG chains to functionalize the outer surface of liposomes (Figure 1C) [41]. Furthermore, a special type of liposome called ultradeformable vesicles was first introduced by Cevc and Blume in 1992 and designated as Transfersomes [42], comprising unilamellar vesicles containing a surfactant responsible for their deformability (Figure 1D). These nanocarriers were specially developed for dermal and transdermal drug delivery. 

Liposome characteristics depend on the lipid composition selected for their preparation. Accordingly, they can present different surface charges: neutral, positive or negative. Non-charged and non-coated liposomes tend to be less stable and higher aggregation is expected. On the opposite, charged liposomal formulations give rise to an increase in the intermembrane repulsion and reduction in the propensity of liposomes to aggregate.

Negatively charged vesicles present enhanced interaction with the MPS and were used in pathologies localized in the liver and spleen [43]. Cationic liposomes were primarily developed for gene delivery, as negatively charged nucleic acids electrostatically interact with positively charged vesicles allowing cell transfection. Other applications can be to deliver chemotherapeutic drugs to formed tumor blood vessels or to pass through the blood–brain barrier for effective drug delivery into the central nervous system [44].

Another targeting strategy that has been developed is the preparation of stimuli-responsive liposomes, i.e., liposomes can be designed to release the incorporated drugs in the presence of specific stimuli, such as an in vivo pathological trigger. The stimuli can be tissue-generated (pH or redox potential) or generated from external sources (temperature, ultrasound or electromagnetic field) [45].

The phospholipids traditionally used for liposomes preparation are phosphatidyl choline from egg yolk or soybean, hydrogenated phosphatidyl choline, phosphatidyl ethanolamine, phosphatidyl glycerol, phosphatidyl inositol, phosphatidyl serine and phosphatidic acid [46]. The modification of the nonpolar and polar regions leads to the synthesis of more stable phospholipids than the respective natural forms [47]. The composition of liposomes can be also modified by several components other than PEG such as poly(aminoacids), heparin, dextran and chitosan, to replace PEG, and antioxidants and complexing agents, to increase chemical stability. Additionally, cyclodextrins, for incorporation of hydrophobic molecules inside the vesicle aqueous core can be successfully associated [47].

There are several types of liposome classifications, the most common according to the structural parameters, method of preparation or with their behavior in vivo. Liposomes are typically classified by the number of lipid bilayers (lamellae) (which can be uni, multi or plurilamellar). Vesicles are characterized by the diameter (which may vary from a few nanometers to a few micrometers) and distribution of vesicle populations (assessed by the polydispersion index); captured volume (aqueous volume sequestered by amount of lipid); drug loading or drug incorporated per amount of lipid in liposomal form; incorporation/encapsulation efficiency, expressed as a percentage and corresponds to the quotient between the final and initial ratios of incorporated substance/lipid in liposomal form; surface charge; and morphology [8,48].

The classic method of preparing liposomes is extremely simple, consisting of dissolution of the mixture of the lipid constituents (containing the bioactive agent if hydrophobic) in compatible organic solvent, followed by drying in a round bottom flask, under reduced pressure or under a flow of nitrogen, to form a lipid film. The addition, under stirring, of an aqueous solution (containing the bioactive agent in the case of being hydrophilic) leads to the formation of liposomes in suspension. This method generates multilamellar liposomes containing the incorporated agent, commonly referred to as multilamellar vesicles (MLV). Despite its simplicity, this method of preparation has some drawbacks, as resulting liposomes present great heterogeneity in terms of mean size and low incorporation efficiencies. Small unilamellar vesicles, commonly referred to as SUV can be obtained by submitting large liposomes, such as MLV, to the action of ultrasound or extrusion. Large unilamellar vesicles (LUV), can be achieved after solubilization of the lipid mixture and of the bioactive agent in detergent and its slow removal by dialysis. All these types of liposomes, MLV, LUV and SUV, are named as conventional or from the first generation, and were progressively being replaced by more sophisticated ones that allowed the incorporation of higher percentages of solutes. One of these preparation strategies involves a lyophilization step of liposomes, containing the incorporated material, and rehydration, under controlled conditions. This kind of liposomes, called dehydration–rehydration vesicles (DRV) represent one of the first important stages in liposome technology, and allows the incorporation efficiency to increase from around 10% in classical liposomes up to 90% [49]. A completely different method, commonly referred to as “active loading”, allows the incorporation of bioactive agents in preformed liposomes with full preservation of its integrity. However, this is only applicable to bioactive agents that are weak bases or acids. The extrusion of liposome suspensions obtained from any methods is used for the homogenization of the vesicle size of liposomes [48].

The encapsulation of enzymes in liposomes is a dual strategy aiming to protect enzymes from degradation and to target them in vivo. In the recent years, enzyme entrapment methods have been based on kinetic specifications of the enzyme, and optimum pH and temperature in a way to guarantee that both an enzyme-loaded liposome and a free enzyme showed a typical Michaelis–Menten profile [50]. Additionally, microfluidics technology has been applied to prepare enzyme-loaded liposomes. Recently, microfluidics was used to produce superoxide dismutase-loaded liposomes with high encapsulation efficiency and maintaining the biological activity of the enzyme [51]. This method of preparation presents advantages over conventional preparation methods as the low time-consuming and the easily up-scaling microfluidic assembly method.

## 4. Enzymatic Liposomal Formulations

As already mentioned, the encapsulation of enzymes in DDS has been extensively investigated aiming to improve their therapeutic performance. Liposomes are one of the most versatile nanosystems for the delivery of therapeutic enzymes, either by their ability to modulate the characteristics according to the membrane composition and size or by the technological approaches used for their production. 

Moreover, the development of liposomal formulations with native or chemically modified forms, their performances, the different routes of administration and lipid compositions have been extensively reviewed [8,52,53,54] including the use of mixed lipid deformable vesicles (e.g., Transfersomes) especially designed for transdermal delivery [55,56]. 

Relevant examples of therapeutic enzymes associated with liposomes, particularly, for the most well-studied, *L*-Asparaginase and Superoxide Dismutase, liposomal formulation details and the most important outcomes of pre-clinical assays are presented in Table 3. 

**Table 3 pharmaceutics-14-00531-t003:** Selected pre-clinical studies using enzymatic liposomal formulations.

Enzyme	In Vivo Model	Lipid Composition	Aim of the Study	In Vivo Outcome	Ref.
Acylated *L*-Asparaginase	Lymphoma	EPC: Chol: PI EPC: Chol: SA	Therapeutic validation of acylated *L*-Asparaginase liposomal formulations	Acylated *L*-Asparaginase liposomes: 8-fold increase in half-life (i.v.) without eliciting adverse effects and higher antitumor effect in comparison to free acylated enzyme.	[57]
*L*-Asparaginase	Lymphoma	EPC:Chol:GM1 EPC:Chol:PI EPC:Chol:SA	Evaluation of antileukemic activity of *L*-Asparaginase liposomal formulations	L-Asparaginase Liposomes: 15-fold increase in half-life (i.v.) without eliciting adverse effects, higher antitumor activity and increased survival rate than the free enzyme.	[58]
*L*-Asparaginase	Lewis lung Carcinoma	SPC:Chol:DSPE-PEG	Evaluation of antitumor activity	*L*-Asparaginase liposomes improved the survival rate of mice induced with Lewis lung carcinoma in comparison to the free enzyme	[59]
Catalase	Melanoma	SPC:Chol:DSPE-PEG: DSPE-PEG-NH2-aPDL1	Development of immunoliposomes with specificity to B16F10 cells loaded with catalase to induce tumor hypoxia	Targeted liposomes accumulated at tumor sites; reduced tumor progression and enhanced survival rate of induced animals.	[60]
Superoxide Dismutase	Adjuvant rheumatoid arthritis	EPC:Chol:SA EPC:Chol:PI.	Therapeutic effect: influence of the lipid composition	Superoxide dismutase liposomes (i.v.) displayed higher therapeutic activity than the free enzyme.	[61]
Superoxide Dismutase	Adjuvant rheumatoid arthritis	EPC:Chol:DSPE-PEG	Influence of mean size and route of administration of superoxide dismutase liposomes	i.v. and s.c. injection of low mean size liposomes (110 nm) displayed similar anti-inflammatory effects. i.v. injection of high mean size liposomes (450 nm) exhibited higher anti-inflammatory than s.c.	[62]
Superoxide Dismutase	Adjuvant rheumatoid arthritis	EPC:Chol:SA EPC:Chol:DSPE-PEG	Therapeutic effect of superoxide dismutase liposomes: influence of the lipid composition	Long circulating superoxide dismutase liposomes (DSPE-PEG) (i.v.) displayed the highest therapeutic activity in comparison to all formulations tested.	[63]
Superoxide Dismutase	Adjuvant rheumatoid arthritis	SPC:Sodium cholate	To treat paw inflammation by middleical application of superoxide dismutase on a remote site	Superoxide dismutase liposomes middleically applied reached blood circulation.	[56]
Superoxide Dismutase	Adjuvant rheumatoid arthritis	EPC:Chol:DSPE-PEG EPC:Chol:SA	Biodistribution and therapeutic effect of superoxide dismutase vs. acylated superoxide dismutase liposomes: influence of lipid composition	Higher accumulation and anti-inflammatory activity at affected sites of DSPE-PEG liposomes than SA liposomes (i.v.). Acylated superoxide dismutase liposomes exhibited earlier therapeutic activity than the native enzyme.	[64]
Superoxide Dismutase	Adjuvant rheumatoid arthritis	EPC:Chol:DSPE-PEG:DSPE-PEG-maleimide EPC:Chol:DSPE-PEG:	In vivo performance of superoxide dismutase liposomes vs. superoxide dismutase enzymosomes (covalently linked at DSPE-PEG distal terminus)	Superoxide dismutase enzymosomes even with a decrease in blood circulation times showed earlier therapeutic activity than superoxide dismutase liposomes (i.v.).	[65]
Superoxide Dismutase	Ischemia-reperfusion	EPC:Chol:DSPE-PEG:DSPE-PEG-maleimide EPC:Chol:DSPE-PEG:	In vivo performance of superoxide dismutase liposomes vs. superoxide dismutase enzymosomes (covalently linked at DSPE-PEG distal terminus)	Superoxide dismutase enzymosomes enhanced therapeutic activity as compared to superoxide dismutase liposomes (i.v.).	[66]
Superoxide Dismutase	Ear oedema	EPC:Chol:DSPE-PEG	Anti-inflammatory effect: Influence of route of administration	Higher edema inhibition for superoxide dismutase liposomes (i.v.) vs. free superoxide dismutase.	[67]
Streptokinase	Human clot inoculated rat model	DOPE:c(RGD)	Thrombolytic activity and release of SK from liposomes	SK liposomes exhbited higher thrombolytic activity than the free enzyme.	[68]
Streptokinase	Thromboembolism	DSPC:Chol:DSPE-PEG	Pharmacokinetic of SK liposomes vs. free enzyme	Increasd blood circulation half-life (16-fold higher) for SK liposomes than the free enzyme (i.v.).	[69]
Urokinase	Thromboembolism	DPPC: DSPE-PEG: c(RGD)	Pharmacokinetic studies and binding to activated platelets of urokinase: free vs. liposomal forms	Increased half-life and improved thrombolytic efficacy: 4-fold over the free enzyme (i.v.).	[70]
Uricase	Uric acid reduction	nanosomal microassemblies	Validationt of an efficient and safe formulation of uricase	Increased circulation time, raised bioavailability, and enhanced uric acid-lowering efficacy, while simultaneously decreasing the immunogenicity.	[71]

ALL—acute lymphoblastic leukemia; EPC—egg phosphatidyl choline; Chol—cholesterol; PI—phosphatidyl inositol; SA—stearylamine; GM1—monosialo gangliosides; SPC—soya phosphatidyl choline; DSPE-PEG—distearoyl phosphatidyl ethanolamine covalently linked to polyethylene glycol; DOPE—dioleoyl phosphatidyl ethanol amine; DSPC—distearoyl phosphatidyl choline; DPPC—dipalmitoyl phosphatidyl choline; SK—streptokinase.

*L*-asparaginase is a component of the chemotherapy cocktail for the treatment of acute lymphoblastic leukemia (ALL) since the 1960s, acting by depletion of the circulating substrate, L-asparagine [72]. This gives rise to starvation and apoptosis of leukemia cells as they are unable to synthesize asparagine for their own metabolism [73]. The main drawbacks of asparaginase treatment are due to relevant toxicity, immunogenic reactions and short blood half-life after parenteral administration [74]. As the substrate of the enzyme is circulating in the bloodstream, to maximize asparaginase therapeutic efficacy, long blood circulating times are required. This was achieved in the decade of 1990s following its incorporation in liposomes, with enzymatic activity preservation higher than 99% [75]. Nanosized *L*-Asparaginase liposomes (~150 nm) demonstrated to enhance half-life in the bloodstream (up to 10 times) in comparison to the non-incorporated enzyme. In addition, the incorporation in liposomes was responsible by the increased therapeutic effect observed in a tumor P1534-induced mice model, as well as the absence of acute toxicity [58]. Similar therapeutic performance was also described using a chemically modified L-Asparaginase [76] after incorporation in liposomes [57]. The long blood circulating properties of these two *L*-Asparaginase nanoformulations were accomplished by including in the lipid composition of liposomal membrane certain kind of phospholipids such as monosialogangliosides or phosphatidyl inositol. These lipids present at the liposomal surface are responsible for the decrease in plasmatic protein adsorption and consequent uptake by the MPS [57,58]. More recently, a similar effect of Asparaginase loaded in long circulating liposomes, by including DSPE-PEG in the lipid composition, improved the therapeutic effect, in a Lewis lung carcinoma murine model, to a higher extent in comparison to the enzyme in the free form [59].

Catalase is an oxyreductase enzyme with antioxidant activity acting by degrading hydrogen peroxide produced in inflammatory processes. After being co-loaded in liposomes with a cytotoxic drug, cisplatin, the nanoformulation was able to induce tumor hypoxia, resulting in a synergistic therapeutic effect [77]. In another study performed by Hei and collaborators, catalase was also incorporated in liposomes, to overcome tumor hypoxia, presenting at their surface a programmed death ligand 1 monoclonal antibody to enhance immunotherapeutic effects towards melanoma. This combination therapy resulted in improved liposome accumulation at tumor sites, reduction in tumor progression and enhanced survival rate of induced animals [60].

Superoxide dismutases are a family of metalloenzymes, able to scavenge superoxide radicals thus acting as antioxidants for the treatment of reactive oxygen species-mediate diseases. For more than 30 years, superoxide dismutase has been studied as a therapeutic tool in many animal models as well as in few clinical trials with positive results [53,78,79,80]. Although the superoxide dismutase potential therapeutic effect, a major limitation of its in vivo use is the rapid elimination from blood circulation with a plasma half-life of 6 and 20 min in rats and in humans, respectively [62,81]. Several strategies have been used to prolong the circulation time and consequently to improve the therapeutic outcome of superoxide dismutase namely its chemical modification or association to polymeric or lipid-based systems, namely liposomes [64,78]. In general, the incorporation of superoxide dismutase in liposomes with different properties has resulted in higher therapeutic activity in comparison to the respective free form in several animal models such as the rat model of adjuvant arthritis, ear oedema model and ischemia reperfusion injury [56,61,62,63,64,65,66,67]. Several therapeutic studies of superoxide dismutase delivery by conventional liposomes, described essentially up to 1989 were reviewed elsewhere [82]. In a rat model, after hyperoxia exposure the i.v. injection of conventional liposomes containing superoxide dismutase offered an efficient protection against oxygen toxicity [83]. The incorporation of SOD in conventional liposomes increased therapeutic activity in a rheumatoid arthritis murine model. This in vivo outcome was dependent on the type and mean size of liposomes being potentiated for formulations with higher blood circulating properties [84]. The improvement in pharmacokinetics properties and the enhanced passive targeting capacity to inflamed sites further increased the therapeutic activity of the superoxide dismutase long circulation liposomes as compared to the conventional formulation [63]. The route of administration (i.v. vs. s.c.) and mean size of superoxide dismutase liposomes demonstrated that size is a critical parameter to allow the vesicles to be drained from the s.c. site of administration. In fact, a lower therapeutic effect of superoxide dismutase liposomes s.c. administered with a mean size of 450 nm when compared to the same superoxide dismutase formulation i.v. injected was attained. Such differences were not observed when small size liposomes were used (110 nm). These different in vivo results were correlated to the different doses of superoxide dismutase that reach the bloodstream [62].

Aiming to further improve the therapeutic effect of superoxide dismutase, the conjunction of enzyme chemical modification followed by incorporation in liposomal lipid bilayer was studied [8,78,85,86]. In the rat model of adjuvant arthritis, this kind of enzymatic nanoformulations, enzymosomes, displayed a faster therapeutic effect when compared to the non-modified form of superoxide dismutase liposomal formulation, as the chemically modified enzyme being partially exposed at vesicles surface is able to exert its enzymatic activity without the need of prior release from the nanosystem [64].

Another combined approach consists of covalent linkage of superoxide dismutase to the terminus of the lipid component (DSPE-PEG), achieving long circulating properties of liposomes, while evidencing superoxide dismutase activity at the outer surface of the nanosystem [65]. The presence of the enzyme at the liposomal surface did not compromise the long circulation time (~16 h) of the nanosystem. Due to this particularity, an earlier therapeutic activity in the rat model of adjuvant arthritis was observed in comparison to superoxide dismutase encapsulated in liposomes [65]. More recently, in a rat model of ischemia reperfusion injury (IRI), superoxide dismutase-enzymosomes were more effective in reducing IRI in comparison to superoxide dismutase encapsulated in liposomes as the therapeutic action is required during the first 24 h, which is not possible when superoxide dismutase is encapsulated in the internal aqueous compartment of liposomes unabling its release at such an early time-point [66].

Besides parenteral administration of superoxide dismutase nanoformulations, non-invasive routes of administration have also been studied. One of these examples refers the epicutaneous application of a superoxide dismutase lipid-based nanoformulation on the back of adjuvant-induced arthritis rats. In this case, enzyme-loaded deformable lipid vesicles, Transfersomes, especially designed for transdermal delivery, were able to improve the disease symptoms of the animals in comparison to induced and non-treated animals, namely paw edema and inflammatory hematological and biochemical parameters. It was also shown that superoxide dismutase-loaded Transfersomes, epicutaneously applied, were able to reach blood circulation in a therapeutic dose and accumulate at inflamed paw [56].

In an experimental colitis model, a recombinant form of superoxide dismutase was loaded in negatively charged liposomes and exhibited a reduction in inflammation of the colon [87]. Additionally, Vorauer-Uhl and co-workers applied topically a rh-CuZn-superoxide dismutase liposomal formulation on a deep second-degree burn on albino rabbits and observed a reduction in the skin swelling, preventing a fully necrotic at the zone of stasis when compared with direct rh-CuZn-superoxide dismutase injection into the lesion or by spreading a rh-CuZn-superoxide dismutase gel [88].

The anti-inflammatory activity of superoxide dismutase-loaded, catalase-loaded and co-encapsulated superoxide dismutase/catalase in ultradeformable vesicles was measured after topical application on the arachidonic acid-induced mouse ear oedema model. The biological activity was compared to enzyme-loaded conventional vesicles and enzyme solutions. The co-association of both enzymes to deformable vesicles resulted in an improvement in enzyme penetration across the intact skin barrier to exert their therapeutic effect [55].

Streptokinase is a fibrinolytic protein and urokinase is a serine protease. They are enzymes widely used for the treatment of several thrombosis disorders namely myocardial infarction, acute cerebral infarction and pulmonary embolism. Streptokinase activates bond cleavage to produce plasmin and urokinase acts by catalyzing the production of plasmin, leading to the breakdown of the fibrin mesh structure in blood clots. Their reduced biological half-lives constitute important drawbacks limiting the clinical use. Pharmacokinetic studies using streptokinase loaded in long circulating liposomes resulted in an increase in the half-life of the enzyme (16-fold higher) in comparison to the respective free form [69], and a higher thrombolytic activity than the free enzyme in a human clot inoculated rat model was achieved. The lipid nanoformulation was able to bind effectively to activated platelets and release the payload in a much higher extent than the free enzyme [68]. In a study conducted by Zhang and co-workers, urokinase following its incorporation in a cyclic RGD, functionalized long circulating liposomes led to an increase in the half-life of the enzyme. This nanoformulation evidenced a 4-fold higher thrombolytic activity when compared to urokinase in the free form, in a mouse mesenteric thrombosis model [70].

Uricase is an enzyme that catalyzes the degradation of uric acid being then by-products easily eliminated by renal excretion. As observed for other therapeutic enzymes, the clinical application of uricase has been limited by unsuitable biological properties namely premature degradation and inactivation by endogenous proteases, among others [89]. Following a common strategy described before for other enzymes, uricase incorporation in liposomes resulted in increased half-life, enhanced uric acid-lowering efficacy and decreased immunogenicity over free enzymes [90].

## 5. Conclusions

While enzymes as therapeutic agents have been applied for more than one century, their use as biopharmaceutics is still far from the expected for such a potent class of molecules. This fact has been correlated with side effects and low residence time in the living organisms, drawbacks that need to be solved. Fortunately, on the one hand, the DNA technology advancements allow enzyme production with necessary quantities and appropriated characteristics for clinical use. On the other hand, pharmaceutical nanotechnologies made possible the modulation of enzyme presentation to the human organisms, by its association with DDS namely liposomes, thus improving their therapeutic activity. Besides all the benefits from incorporating therapeutic enzymes in liposomes, the translation from the laboratory to the clinic always needs to address important challenges. One main challenge concerning enzymes is related to the preservation of the protein 3D structures, not only to avoid immunogenic responses but to maintain their biological function. During nanoformulation development, for each enzyme, the lipid composition needs to be adequately selected aiming to maximize the loading and to maintain the enzyme integrity. Furthermore, liposomal membrane compositions must be designed to deliver the enzyme with the suitable biodistribution profile according to the specific organ or tissue that it is intended to target. Another challenge deals with the chemical and physical stability of the liposomal membrane. In fact, liposome stability issues, observed for conventional liposomes, mostly used in the 1980s and 1990s decades, have been overcome due to intense academic efforts, resulting in the accomplishment of several methodologies to achieve stable liposomes. Moreover, the large-scale production of liposomal formulations is a well-established process that has already given rise to several commercialized products. Successful examples in clinical use are more than a dozen liposomal nanoformulations. The continuous evolution and sophistication of liposome technology as well as the increased fields of application are expected to enlarge this clinical offer. Nowadays, the liposome technology is mature, and these DDS will continue to play a leading role in bringing new enzymatic nanomedicines to the market. We truly believe that enzymatic therapy after appropriate formulation in DDS, such as liposomes, will fulfil its due place within the battery of the new generation of biomedicines. 

## Figures and Tables

**Figure 1 pharmaceutics-14-00531-f001:**
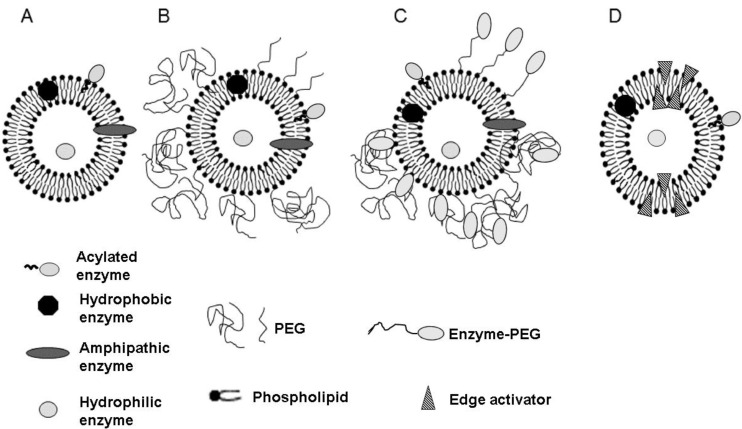
Schematic representation of different types of liposomes: (**A**)—Conventional liposome loaded with hydrophilic enzyme in the internal aqueous space, and hydrophobic, amphipathic and acylated (hydrophilic enzyme linked to fatty acid chains) enzymes in the lipid bilayer; (**B**)—Enzyme-loaded liposomes containing lipids linked to polyethylene glycol (PEG), called long-circulating liposomes; (**C**)—Enzyme-loaded liposomes containing simultaneously PEG and enzymes at liposome surface or linked to functionalized PEG; (**D**)—Enzyme-loaded ultra-deformable liposomes, Transfersomes), with the non-uniformly distributed edge-active components at the stressed sites for membrane deformation. Vesicle D was intentionally drawn in an oval shape to represent the deformity that characterizes it and that is caused by the edge activator molecules. Adapted from Torchilin et al. 2005 [35].

**Table 2 pharmaceutics-14-00531-t002:** Commercialized enzyme-based drugs. Adapted from selected revision articles [12,27].

	Disease/Deficiency	Therapeutic Enzyme Name	Commercial Name
Enzyme Deficiency	Gaucher’s	Alglucerase; Imiglucerase	Ceredase, Cerezyme, Taliglucerase
	Fabry’s	Agalsidase	Febrazym
	Hunter’s	Iduronate-2-sulfatase	Elaprase
	Huler’s	α-*L*-iduronidase	Aldurazyme
	Pompe’s	α-glucosidases	Myozyme
	Sucrase-isomaltase	Sacrosidase	Sucraid
	Immunodeficiency	Pegademase	Adagen
	Morquio syndrome	*N*-acetylgalactosamine-6-sulfate sulfatase	Vimizim
	Maroteaux-Lamy syndrome	*N*-acetylgalactosamine-4-sulfatase	Naglazyme
	Sly syndrome	β-glucuronidase	Mepsevii
	α-Mannosidosis	Velmanase α	Lamzede
	Batten disease	Cerliponase α	Brineura
Circulation and cardiac problems		Nattokinase	Streptase, Syner-Kinase, Kinclytic, Rapilsyn, Actilyse, Metalyse
Several Cancer		Pegylated arginine deiminase	ADI-PEG 20
		Rasburicase	Fasturtec, Elitek
Leukemia		*L*-asparaginase	Spectrial, Kidfrolase, Oscarpar, Erwinase
Cystic fibrosis		Dornase alfa	Pulmozime
Digestive Disorders		Pancreatic enzymes	Theraclec total
		Proteases, lipases, amylases	Several

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
