# Peer review of "Liposomes as Tools to Improve Therapeutic Enzyme Performance"

_pharmaceutics, 2022, doi:10.3390/pharmaceutics14030531_

Round 1

Reviewer 1 Report

The efficient delivery of peptide and proteinaceous drugs is still a challenge and thus the review "Liposomes as tools to improve therapeutic enzymes performance "can be interesting for a wide audience of researches. However, the well-known drawback of liposomes as drug delivery vehicles is their low stability, but the authors did not address the stability issue at all in their review. Could the authors, please, provide information on liposome advantages/drawbacks, limitations, perspectives and challenges in future development of liposome technology in relation to enzyme delivery?

Line 324-327 "In general, the incorporation of superoxide dismutase in liposomes with different properties has resulted in higher therapeutic activity in comparison to the respective free form in several animal models such as rat model of adjuvant arthritis, ear oedema model, ischemia reperfusion injury" – References are needed here.

Author Response

  1. The efficient delivery of peptide and proteinaceous drugs is still a challenge and thus the review "Liposomes as tools to improve therapeutic enzymes performance "can be interesting for a wide audience of researches. However, the well-known drawback of liposomes as drug delivery vehicles is their low stability, but the authors did not address the stability issue at all in their review. Could the authors, please, provide information on liposome advantages/drawbacks, limitations, perspectives and challenges in future development of liposome technology in relation to enzyme delivery?

Reply: Thank you for your suggestion. In fact, we have not addressed drawbacks, limitations and lack of stability of the liposomes/enzymes in the manuscript. We are providing now in the conclusions section the advantages/drawbacks, limitations, perspectives, and challenges in future development of liposome technology.

In the revised version of the manuscript the following sentences were included:

“Besides all benefits from incorporating therapeutic enzymes in liposomes, the translation from laboratory to clinic always needs to address important challenges. One main challenge concerning enzymes is related to the preservation of the protein 3D structures, not only to avoid immunogenic responses but to maintain their biological function. During nanoformulation development, for each enzyme, the lipid composition needs to be adequately selected aiming to maximize the loading and to maintain the enzyme integrity. Furthermore, liposomal membrane compositions must be designed to deliver the enzyme with the suitable biodistribution profile according to the specific organ or tissue that is intended to target. Another challenge deals with chemical and physical stability of liposomal membrane. In fact, liposome stability issues, observed for conventional liposomes, mostly used in the 80´s and 90´s decades, have been overcome due to intense academic efforts, resulting in the accomplishment of several methodologies to achieve stable liposomes. Moreover, the large-scale production of liposomal formulations is a well-established process that has already given rise to several commercialized products. Successful examples in clinical use are more than dozen liposomal nanoformulations. The continuous evolution and sophistication of liposomes technology as well as the increased fields of application are expected to enlarge this clinical offer. Nowadays, the liposome technology is mature, and these DDS will continue to play a leading role in bringing new enzymatic nanomedicines to the market. We truly believe that enzymatic therapy after appropriate formulation in DDS, such as liposomes, will fulfil its due place within the battery of the new generation of biomedicines. “

  1. Line 324-327 "In general, the incorporation of superoxide dismutase in liposomes with different properties has resulted in higher therapeutic activity in comparison to the respective free form in several animal models such as rat model of adjuvant arthritis, ear oedema model, ischemia reperfusion injury" – References are needed here.

Reply: In fact, the references were missing. The references were included in the revised version of the manuscript:

“In general, the incorporation of superoxide dismutase in liposomes with different properties has resulted in higher therapeutic activity in comparison to the respective free form in several animal models such as rat model of adjuvant arthritis, ear oedema model, ischemia reperfusion injury [56, 61-67].”

Reviewer 2 Report

The present review entitled "Liposomes as tools to improve therapeutic enzymes performance" reviews the therapeutic potential of enzymes loaded liposomes. The review covers many aspects of enzymes, liposomes and liposomal formulations of enzymes.  However, the manuscript is poorly written and many sentences are meaningless. It needs an extensive editing of the Englsih language. For example: Line 34-36, The sentence is unclear and needs editing. 

The authors need to include an important reference related to therapeutic applications of superoxide dismutase. 

Therapeutic potentials of superoxide dismutase. Int J Health Sci (Qassim). 2018 May-Jun;12(3):88-93. PMID: 29896077

Author Response

  1. The present review entitled "Liposomes as tools to improve therapeutic enzymes performance" reviews the therapeutic potential of enzymes loaded liposomes. The review covers many aspects of enzymes, liposomes and liposomal formulations of enzymes.  However, the manuscript is poorly written and many sentences are meaningless. It needs an extensive editing of the Englsih language. For example: Line 34-36, The sentence is unclear and needs editing. 

Reply: The sentenceBioactive agents include various types of proteins (i.e., enzymes, antigens, antibodies), genetic materials (i.e., DNA, mRNA, siRNA, oligonucleotides), among others” was changed to:

“Some examples of those molecules used as bioactive agents include various types of proteins (i.e., enzymes, antigens, antibodies), genetic material (i.e., DNA, mRNA, siRNA, oligonucleotides), amongst others.” in the revised version of the manuscript.

In addition, the English was revised in the submitted manuscript.

  1. The authors need to include an important reference related to therapeutic applications of superoxide dismutase. 

Therapeutic potentials of superoxide dismutase. Int J Health Sci (Qassim). 2018 May-Jun;12(3):88-93. PMID: 29896077

Reply: The reference suggested by the reviewer was included in the revised version of the manuscript.

Therapeutic potentials of superoxide dismutase. International journal of health sciences12(3), 88–93.

Reviewer 3 Report

The review manuscript entitled "Liposomes as tools to improve therapeutic enzymes performance" involved the interesting bibliographical research of liposomes formulations (DDS) to enhance two selected enzymes (L-asparaginase and superoxide dismutase) performance to treat different pathologies.

Moreover, the authors analyzed the current state-of-art regarding the commercial enzyme-based drug systems.

The introduction is according to the developed topic of the manuscript, and it has updated bibliographical references to support the research.

Moreover, the manuscript is clear, organize, and focused on the topic that is of growing interest due to the potential biological and medical applications.

Additionally, the information they authors described is supported with clear and logical images/figures/tables that summarize all the required data.

Also, several nanocarriers were discussed with emphasis in liposomes carriers and their properties. According to this point, it should be important to report not only the advantages but the disadvantages of these kind of specific carriers (from a critical point of view). (Section 3.1.)

As a suggestion, Figure 1 should be containing the appropriate font sizes and the images design should be identical for each system (the D image was deformed and is displayed in an oval shape). I encourage the authors to improve the image quality of Figure 1 by using different colours for each component of the liposomal system.

Furthermore, the liposomes formulations were analyzed, and their charges were reported for different systems. Although, it is necessary to explain the advantages and disadvantages of each kind of charge (neutral, positive, and negative) due to the nature of this specific review analysis.

Furthermore, I encourage the authors to check some mistakes (in yellow, pdf file attached) such as:

  • Please check the appropriate format for the authors affiliations
  • Line 26, Table 1, Table 2, and throughout the manuscript: L-asparaginase (L should be in italics)
  • Please remember that the unit and the number (e.g. 20 nM – 50 °C – 10 min) should have a blank space between them but is the opposite when the percentage symbol is used (90%)
  • Table 2: N-acetyl…….N should be in italics
  • The use of capital letters; please check and correct several typo mistakes, thanks.

Finally, I would like to invite the authors to add the abbreviation list of words at the end of this manuscript.

I recommend the acceptance of this manuscript after the authors performed the suggested corrections/additions.

Author Response

The review manuscript entitled "Liposomes as tools to improve therapeutic enzymes performance" involved the interesting bibliographical research of liposomes formulations (DDS) to enhance two selected enzymes (L-asparaginase and superoxide dismutase) performance to treat different pathologies.

Moreover, the authors analyzed the current state-of-art regarding the commercial enzyme-based drug systems.

The introduction is according to the developed topic of the manuscript, and it has updated bibliographical references to support the research.

Moreover, the manuscript is clear, organize, and focused on the topic that is of growing interest due to the potential biological and medical applications.

Additionally, the information they authors described is supported with clear and logical images/figures/tables that summarize all the required data.

  1. Also, several nanocarriers were discussed with emphasis in liposomes carriers and their properties. According to this point, it should be important to report not only the advantages but the disadvantages of these kind of specific carriers (from a critical point of view). (Section 3.1.).

Reply: Thank you for the suggestion, however, also as proposed by reviewer 1, we have addressed the advantages/disadvantages limitations, perspectives, and challenges in future development of liposome technology in the conclusions section.

In the revised version of the manuscript the following sentences were included:

“Besides all benefits from incorporating therapeutic enzymes in liposomes, the translation from laboratory to clinic always needs to address important challenges. One main challenge concerning enzymes is related to the preservation of the protein 3D structures, not only to avoid immunogenic responses but to maintain their biological function. During nanoformulation development, for each enzyme, the lipid composition needs to be adequately selected aiming to maximize the loading and to maintain the enzyme integrity. Furthermore, liposomal membrane compositions must be designed to deliver the enzyme with the suitable biodistribution profile according to the specific organ or tissue that is intended to target. Another challenge deals with chemical and physical stability of liposomal membrane. In fact, liposome stability issues, observed for conventional liposomes, mostly used in the 80´s and 90´s decades, have been overcome due to intense academic efforts, resulting in the accomplishment of several methodologies to achieve stable liposomes. Moreover, the large-scale production of liposomal formulations is a well-established process that has already given rise to several commercialized products. Successful examples in clinical use are more than dozen liposomal nanoformulations. The continuous evolution and sophistication of liposomes technology as well as the increased fields of application are expected to enlarge this clinical offer. Nowadays, the liposome technology is mature, and these DDS will continue to play a leading role in bringing new enzymatic nanomedicines to the market. We truly believe that enzymatic therapy after appropriate formulation in DDS, such as liposomes, will fulfil its due place within the battery of the new generation of biomedicines. “

  1. As a suggestion, Figure 1 should be containing the appropriate font sizes and the images design should be identical for each system (the D image was deformed and is displayed in an oval shape). I encourage the authors to improve the image quality of Figure 1 by using different colours for each component of the liposomal system.

Reply: Figure 1 is merely a schematic representation of the liposomal systems used for enzyme delivery. The D system represents a Transfersome and it was intentionally done with an oval shape to demonstrate the deformability of the system. The inventors of this type of liposome called it as a mixed lipid aggregate, since the elements responsible for the deformability and adaptability of the system are not fixed in the lipid bilayer, but rather have a random disposition, accumulating in the deformation regions when they are applied to the skin non-occlusively [please see https://doi.org/10.1016/S0005-2736(02)00401-7]. To make the interpretation clearer, we have inserted additional information on vesicle shape to the figure legend.

Figure 1. Schematic representation of different types of liposomes: A - Conventional liposome loaded with hydrophilic enzyme in the internal aqueous space, and hydrophobic, amphipathic and acylated (hydrophilic enzyme linked to fatty acid chains) enzymes in the lipid bilayer; B – Enzyme-loaded liposomes containing lipids linked to polyethylene glycol (PEG), called long-circulating liposomes; C – Enzyme-loaded liposomes containing simultaneously PEG and enzymes at liposome surface or linked to functionalized PEG; D – Enzyme-loaded ultra-deformable liposomes (Transfersomes), with the non-uniformly distributed edge-active components at the stressed sites for membrane deformation.  Vesicle D was intentionally drawn in an oval shape to represent the deformity that characterizes it and that is caused by the edge activator molecules. Adapted from Torchilin et al. 2005 [35].

  1. Furthermore, the liposomes formulations were analyzed, and their charges were reported for different systems. Although, it is necessary to explain the advantages and disadvantages of each kind of charge (neutral, positive, and negative) due to the nature of this specific review analysis.

Reply: We have changed the text to meet the requirements. So, the following text was added in line 194:

“Liposomes characteristics depend on the lipid composition selected for their preparation. Accordingly, they can present different surface charge: neutral, positive or negative. Non-charged and non-coated liposomes tend to be less stable and higher aggregation is expected. On the opposite, charged liposomal formulations give rise to an increase on the intermembrane repulsion and reduction on the propensity of liposomes to aggregate. “

  1. Furthermore, I encourage the authors to check some mistakes (in yellow, pdf file attached) such as:

Please check the appropriate format for the authors affiliations

Reply: Thank you for pointing out all those mistakes and spelling errors. Now, we think that all are corrected. However, we are sorry, but the affiliation of the authors is correct. In line 26, Table 1, Table 2, and throughout the manuscript: in L-asparaginase (L is in italics, The percentage symbol do not have a blank space

In Table 2,: N-acetyl the N  is in italic and  the use of capital letters was checked in all manuscript and corrected.

Finally, an abbreviation list was included at the end of the manuscript.